# Integrated Genetic and Omics Approaches for the Regulation of Nutritional Activities in Rice (*Oryza sativa* L.)

Muhammad Junaid Zaghum [1,2,3], Kashir Ali [4] and Sheng Teng [1,3,*]

1   Laboratory of Photosynthesis and Environmental Biology, CAS Center for Excellence in Molecular Plant Sciences, Institute of Plant Physiology and Ecology, Chinese Academy of Sciences, Shanghai 200032, China
2   Seed Physiology Laboratory, Department of Agronomy, University of Agriculture Faisalabad, Faisalabad 38000, Pakistan
3   University of Chinese Academy of Sciences, Beijing 100049, China
4   Institute of Soil and Environmental Sciences, University of Agriculture Faisalabad, Faisalabad 38000, Pakistan
*   Correspondence: steng@cemps.ac.cn

**Abstract:** The primary considerations in rice (*Oryza sativa* L.) production evoke improvements in the nutritional quality as well as production. Rice cultivars need to be developed to tackle hunger globally with high yield and better nutrition. The traditional cultivation methods of rice to increase the production by use of non-judicious fertilizers to fulfill the nutritional requirement of the masses. This article provokes nutritional strategies by utilization of available omics techniques to increase the nutritional profiling of rice. Recent scientific advancements in genetic resources provide many approaches for better understanding the molecular mechanisms encircled in a specific trait for its up- or down-regulation for opening new horizons for marker-assisted breeding of new rice varieties. In this perspective, genome-wide association studies, genome selection (GS) and QTL mapping are all genetic analysis that help in precise augmentation of specific nutritional enrichment in rice grain. Implementation of several omics techniques are effective approaches to enhance and regulate the nutritional quality of rice cultivars. Advancements in different types of omics including genomics and pangenomics, transcriptomics, metabolomics, nutrigenomics and proteomics are also relevant to rice development initiatives. This review article compiles genes, locus, mutants and for rice yield and yield attribute enhancement. This knowledge will be useful for now and for the future regarding rice studies.

**Keywords:** genome selection; rice breeding; genetic analysis; omics-assisted markers; nutritional quality; genomics and pangenomics; biofortification

## 1. Introduction

Rice grain is an edible and nutritional staple food among cereals in Asian tropical regions [1]. Rice is cultivated on 162 million hectares of land in tropical and subtropical climatic zones having variable thermal regime as aerial temperature range (25 °C to 35 °C) and produces 755 million tonnes annually (FAOSTAT) (http://www.fao.org/faostat/en/#data/QC/visualize (accessed on 10 March 2022). Rice grain contains a variety of complex carbohydrates, amino acids, minerals, nutritional fiber, and vitamins. Due to its use as a staple food in numerous impoverished nations, it offers around 27% of calories, 20% of protein, and 715 kcal/day in the diet. Global population growth has necessitated a double increase in agricultural output and quality to fulfill the rising food demand. Approximately 100 million tonnes of additional rice are estimated to be needed to sustain the world population growth. It is pertinent to mention that despite the level of rice production, enriched nutritional profiling is considered the main domain, which is direly necessary and provokes various health issues, keeping in view food security concerns [2]. The World Health Organization (WHO) establishes standards for the basic composition and structural quality characteristics of rice, depending on the amino acids, mineral, flavonoid content,

proteins, carbohydrate, and essential vitamins that are present in rice grain [3]. Rice has acceptable levels of inorganic and organic unwanted matter, and it is free from toxic heavy metals such as mercury (Hg), arsenic (As), lead (Pb), and cadmium (Cd), thus making it considered to be of edible quality [4]. Rice has plenty source of nutrients, including iron (Fe), calcium (Ca), phosphorus (P), potassium (K), sodium (Na), and essential vitamins [5]. Several methods are adopted to reduce the Cd accumulation in rice by altering the cropping pattern, phyto-remediation, and breeding of Cd-tolerant exclusion in rice fields as root-specific traits [6]. Brown rice is high in iron, proteins, and phosphorus, and it is usually regarded as the nutritious variety of rice due to its good capacity for the uptake and assimilation of nutrients. The famous basmati rice includes 6 g of lipid, 19 g of protein and 364 kcal of energy as compared to jasmine rice, which has 356 kcal of energy, 6.67 g of protein, and an amount of lipids [5].

The FAO has estimated that over two billion people globally suffer from hidden hunger. Developing and impoverished nations have been trying for several years to improve productivity and fulfill the need for some staple crops that provide the majority of nutritional benefits and required calories. This type of usage and evaluation of nutritional quality is missing in other micronutrients, vitamins, and mineral-containing crops [7]. Rice biofortification is one of the most successful strategies in this domain. Rice grains can be biofortified by agronomic techniques, traditional breeding and modern genetic modification in the function of a specific gene to up- or down-regulate for efficient nutrient assimilation and compartmentalization. Conventional plant breeding refers to the process of selecting and crossing desirable characteristics in crops over several generations, while modern biofortification technologies, such as genetic engineering, offer a precise way of altering genetic sequences to confer the desired characteristics in a short time [8]. According to the WHO, the most essential challenge is to fortify rice to eliminate malnutrition among a worldwide population suffering from folic acid, iron and vitamin A deficiencies [3]. Biofortification by genetic advancement is a reliable and cost-effective method for the increasing trend of fortified food for economically deprived groups in the population [8].

The traditional breeding approaches are not very complicated because they are not only dependent on the basic gene pool, which is the only genetic resource and makes significant contributions to rice production and development. In addition, the utilization of transgenic approaches helps in the identification of genetic materials from multiple living organisms that enable the engineering of desired features in plants [9]. Golden Rice is one of the most noticeable achievements of the transgenic method, especially in terms of nutritional quality improvement. Transgenic Golden Rice is a low-cost solution to tackle vitamin A deficiency. Golden Rice has been biofortified with β-carotene, which the human body transforms into vitamin A after digestion. Vitamin A is essential for boosting immunity and preventing night blindness. UNICEF estimates that around 125 million children worldwide are vitamin A deficient. Because rice is eaten globally, it was considered as a source of vitamin A and a transporter for β-carotene. Golden Rice has enormous potential associated with the precise regulation of Genetically Modified Organisms (GMOs) [10]. The highlighted issues curtail the deficiency of vitamin A, and Bangladesh adopted this GMO Golden Rice as a commercial cultivar to curb the vitamin A issue in the nation. Likewise, many technologies have been developed by the CRISPR–Cas gene editing technique and have demonstrated enormous potential in helping to lower sociocultural stigma associated with GMOs. Gene editing techniques are drawn from mechanisms found in living species, including DNA repair and the defensive mechanisms adopted by bacteria against infections, and do not need the insertion of foreign genes into the desired genome. Effective research into genomic technologies is contributing to overcoming the normal barriers to the commercialization of genetically engineered crops [11]. Omics-assisted breeding of rice has the potential for genetic enhancement and is largely considered as a sustainable, suitable, safe, unbiased, and significant method for the betterment of rice crops. The advanced omics techniques encompass a variety of technologies, including proteomics, transcriptomics, nutrigenomics, ionomics, metabolomics, and genomics, that facilitate scientists to predict, recognize, and

analyze a wide variety of reproductive molecules present in a living organism, including RNA, protein, ions, DNA and metabolites [12]. Due to advancements in sequencing techniques during the last decade, a large amount of data of transcriptome, sequence, and whole genome have been created for major crops. The crux of these research endeavors may help the industry stakeholders and surely develop a solution to the problems being faced, in terms of fortification of nutritional quality and minerals, by better utilization of omics approaches in the improvement of rice grain quality.

## 2. Nutritional Quality Enhancement by QTL Mapping in Rice

Over the last few decades, tremendous progress has been made in increasing food output and affordability for resource-poor populations. Milled rice is composed of starch, that is, a complex carbohydrate. Rice has a slightly lower protein content than wheat, maize, and pulses, with protein being the second most essential component of cereal crops, even though little effort has been made to increase [13]. Therefore, fortification of rice with antioxidants, vitamins, modified starch, and dietary fibers are desirable characteristics to make rice a better complete staple meal at a reasonable price [8]. Genetic enhancement of such characteristics needs a thorough study of the genetic control of the trait, the genetics and molecular pathways underlying trait regulation, as well as environmental influences. The majority of these characteristics are complicated and are regulated by a large number of moderate-impact genes. The self-pollinated nature of rice enables the establishment of a variety of mapping populations, which includes F2 population, doubled haploid (DH), backcross inbred lines (BILs) and recombinant inbred line (RIL) [14]. Some newly formed innovative mapping populations such as NAM (nested association mapping) [15] and MAGIC (multiparent advanced generation intercross) populations are used to map complex traits [16].

Mapping of quantitative trait loci was investigated to determine the genetic region controlling rice nutritional quality characteristics. The rice mutants with high Fe and Zn concentrations showed Zn concentrations ranging from 15.36 to 28.95 mg/kg and Fe concentrations ranging from 0.91 to 28.10 mg/kg [17]. The complexities of nutrition quality-related characteristics vary significantly; for example, certain variables, such as folate content, have a limited number of significant QTLs, while others, such as protein content, have a large number of minor-effect QTLs. Considerable effort is being made to identify quantitative trait loci for the protein content of rice grains that are mainly located on rice chromosome segments 3 and 5 [18]. Numerous characteristics associated with nutritional quality are linked, and their QTLs typically co-localize. For example, on rice chromosome 6, retrogradation, peak viscosity, QTLs for gel consistency, amylose concentration, breakdown viscosity, final viscosity, setback viscosity and trough viscosity were identified [19]. Consequently, many improved nutritional traits are adversely controlled, making simultaneous improvement difficult. For example, grain iron content is inversely linked to grain production per plant. Similarly, the phytic acid concentration of rice affects mineral bioavailability [17].

Quantitative trait loci (QTLs) for Fe- and Zn-related characteristics from interspecific and intraspecific crosses have been documented in rice grain and less utilized in molecular breeding for this trait [20,21]. Bi-parental mapping populations are time consuming, expensive and yield a lower quality than association mapping [22]. Association mapping relies on linkage disequilibrium or differences seen in wild or cultivated species and confer a relationship between molecular marker and grain Fe and Zn contents in rice grain and heterogeneity for both traits [23]. QTL and association mapping can be utilized in rice for a variety of traits such as grain yield and attributes, seedling low temperature tolerance, cold tolerance at booting stages, heat-stress tolerance, grain quality-related traits, salinity tolerance, drought tolerance, seedling vigor and grain protein constituents [24].

The increase in QTL span and uncertainty in localization that occurs in the complication of QTL mapping applicability to the breeding program. Consensus QTLs are selected for meta-QTL analysis, and a couple of previously completed research studies are used to

improve the locations of the aforementioned QTLs. Additionally, MetaQTLs are specified at the 95% confidence level [25]. In this perspective, the MetaQTL method offers an ideal chance to combine published QTL mapping information from several studies to determine more exact statistically significant levels and phenotypic changes in rice, as well as accurately characterize the QTL span. On chromosome five, one such example involves a shared QTL for phosphate and phytate [26]. Further study attempts found three MetaQTLs associated with higher Zn and Fe concentrations in rice. A similar MetaQTL study was performed to discover potential genes for salt tolerance, rice root shape, and grain size [27]. Although this research may have a synergistic as well as antagonistic impact of multiple QTLs in enhancement of nutritional characteristics, further studies are direly necessary to explore this aspect in an accurate way [17]. There are a number of genes that regulate nutritious functionality features that have been the focus of substantial research in recent years (Table 1). Genes are being efficiently explored with molecular breeding, transgenic method, and even comparatively new technology like genome-editing

**Table 1.** Some rice genes that control regulation of nutritional quality traits.

| Gene | Function | Locus | References |
|---|---|---|---|
| glu4a | Gene involved in storage proteins of seed | Os01g55690 | [28] |
| lpa1 | Metabolizes the phytic acid | Os02g57400 | [29] |
| OsbZIP58, OsSMF1 | Helps in accumulating the storage protein | Os07g08420 | [30] |
| OsNAS3 | Improves the fortification of iron in rice seed | Os07g48980 | [31] |
| OsVIT2 | Involved in translocation of iron | Os09g23300 | [32] |
| OASA2 | Synthesis and accumulation of Tryptophan | | [33] |
| OsYSL2 | Transportation of manganese and iron at long distance | Os02g43370 | [34] |
| RAG2 | Functioning in yield and grain quality | Os07g11380 | [35] |
| LRP, RLRH1, and RLRH2 | Accumulation of lysin content | | [36] |
| XS-lpa2-1 | Involved with phytic acid accumulation | Os03g04920 | [37] |
| TKTKK1 and TKTKK2 | Regulation and synthesis of Methionine and cysteine | | [38] |
| AtGTPCH | Synthesis folate | | [39] |

## 3. GWAS Analysis Improves Rice Nutritional Quality Traits

While the effectiveness in identifying segments of chromosome linked to characteristic QTL mapping has two significant downfalls, the QTL mapping resolution is very limited and is only used to study segregated alleles from the parent line [40]. These drawbacks of QTL mapping are eliminated by using the GWAS techniques [41]. GWAS is a technique for rapidly scanning markers throughout the whole set of DNA to identify genetic changes linked to a certain trait of several species. Following the finding of novel genetic interactions, efficient breeding methods can be used to enhance the yields of rice and other crops [42]. Additionally, the GWAS technique has numerous drawbacks, including the increase in genotype markers, diverse resources of a large germplasm collection, and allele data, such as the presence of minor alleles in at least 5% of the germplasm pool [41]. For unique alleles found in a small number of genotypes, QTL mapping is generally the best method. Recognizing the limits of both methods, it is recommended that they be used in conjunction to identify QTL [43]. The rice seed-related characteristics are analyzed by the usage of QTL mapping and GWAS analysis in combination. The concentrations of Zn, Mo, As, and

Cu in 300 brown rice varieties were determined by using GWAS mapping [44]. These elements show variations in grain composition that are linked to the number of candidate genes and SNPs, and the main reason for variation is environmental circumstances [45]. The mappings of GWAS and QTL are performed in combination to investigate traits such as Al accumulates, although combined research on grain nutrient content is rare [46]. In comparison to QTL mapping, fewer efforts are undertaken to characterize nutritional quality traits using GWAS.

## 4. Efficient Nutrient-Rich Rice Breeding through Genome Selection (GS)

Molecular marker-assisted breeding is an effective strategy for incorporating desirable characteristics from a pool of high-yielding cultivars, which can only be performed with previous information on specific gene loci, associated markers, and repeated backcrossing of large segregating progenies [13]. Additionally, the recently introduced trait might not always improve as predicted, because it belongs to a diverse genetic background, and the undesired attachment leads to significant issues with marker-supported breeding [47]. [48] proposed genomic selection (GS) to overcome these constraints by estimating the potential of breeding lines of rice that are based on high-density markers and phenotypic values. Genome selection is a genetic analysis that is performed by using marker selection, in which the genetic markers of whole genome are applied to ensure the linkage of QTL with at least one marker [49]. Genome selection is being reconsidered in light of current genotyping technologies such as genotyping of the next generation. The efficacy of genome selection analysis is enhanced and made cost effective by innovative genotyping methods [50]. Despite the availability of several genotyping technologies and whole-genome-sequenced genotypes, the genome selection method takes relatively more effort for rice [51]. Genome selection is more likely to be utilized in the addition of NGS (next-generation selection) genotyping technologies in many breeding processes. The GS genotyping technique is cost effective and it increases the efficacy of genome selection technology many times. There are many genotyping methods that are publicly introduced, such as whole-genome sequences, but the usage of the genome selection method for rice genotyping is performed with minimum effort [52]. The efficacy of genome selection was studied in rice for the first time by using inbred lines to improve grain or seed quality characteristics such as height of the plant, total yield, grain yield, and blooming duration [51]. It was discovered through the combination of GS and GWAS that genomic forecast models outperformed pedigree-based prediction in predicting the phenotype [53]. This study shows that the expense of genotyping technology has increased the value of GS, and when coupled with GS and GWAS data on genetic layout and population size, rice breeding efficiency is also boosted [54]. In comparison to yield-related features, the majority of quality-related traits may be predicted accurately. Because quality characteristics have a greater heritability, implementing GS becomes easier [50].

Many other research findings on the assessment of colored rice for various vitamins, antioxidant compounds and minerals have noticed considerable variation, and these accessions were found to contain three to four times more nutrients than advanced rice varieties [55]. In a research study, 30 (53%) quantitative trait loci were co-located with identified or functionally related genes. OsZFP252, OsHMA9, OsNRAMP7, OsMAPK6, and OsMADS13 were among the significant candidate genes for grain Zinc (Zn). Sayllebon, a red rice genotype that is high in both anthocyanins and zinc, could be a valuable breeding material for nutritious rice. QTLs may be utilized for both QTL pyramiding and genomic selection. Some of the discovered QTLs may be validated further by detailed mapping and functional characterization [56].

A genome-wide association study (GWAS) was conducted, and 29 marker–trait associations (MTAs) with significant relationships for characteristics, including ZnMR (5 MTAs), FeBR (6 MTAs), FeMR (7 MTAs) and ZnBR (11 MTAs) [57]. The co-localization of the MTAs controlling the linked features indicates the prospect of their improvement throughout. The associated robust MTAs could be a valuable source of information for enhancing Fe

and Zn concentration in rice grain and addressing Fe and Zn malnutrition among rice consumers [58].

The GEBV (genomic estimated breeding values) computed using the GS technique demonstrated a broad range of reliability for characteristics within the rice plant such as flowering time, plant height, grain yield, and panicle weight. To understand the impact of population structure and marker density on the reliability of genomic prediction, researchers may also look at the structure of characteristics, as well as the reliability of prediction based on genotype [59]. In 2014, a novel approach termed genomic hybrid breeding for the prediction model was suggested with the combination of epistasis and dominance [60], it being a combination of phonological modeling and genome prediction to enhance the phenotypic prediction of complex traits among various settings for genomic hybrid breeding of rice [61]. While genome selection is increasingly utilized to examine rice quality features, the investigation into its efficacy in evaluating nutritional aspects remains lagged.

## 5. Mutation Mapping and Mutagenesis Techniques: Impact on Nutritional Quality of Rice

Mutations contribute to heredity and genetic diversity and are utilized to investigate the functioning of several genes. Conventional hybridization utilizes known genetic changes, and new mutations are occasionally added to acquire unique characteristics. Mutations can occur spontaneously or be caused by chemical and physical agents: chemical agents such as ethyl methane sulfonate and diepoxybutane; physical agents such as gamma rays, fast neutrons, thermal neutrons, UV light, X-rays, and beta and alpha particles; intercalating agents such as ethidium bromide; alkylating agents such as ethylmethanesulfonate [62].

The combination of hybridization and gamma rays have been utilized to produce novel varieties of aromatic rice that increase the level of iron content, indicating that hybridization methods can be used to generate new cultivars with much better characteristics. Numerous mutant rice lines have been produced, including the Thai jasmine rice, in which the anthocyanin concentration was enhanced to give it a blue color by ion-beam bombardment [63]. The BKOS jasmine rice variety is extracted from the mutant strain that enhanced the antioxidant activity and had high phenol content [64]. The mutant lines of low phytic acid (LPA) were produced by using physical and chemical mutagenesis from the Japonica rice and Indica rice species, because phytic acid is usually recognized as an antinutritional factor [65].

While identifying and inducing mutations takes significantly less effort and is easier, the future use of mutation for the breeding process requires more considerable work. Many approaches for mutation mapping are established with the introduction of the next-generation sequencing technique (NGS) [66]. For example, MutMap is a new technique used for identifying primary mutation mapping and mutant loci through the use of sequencing variation in segregated mutant lines. In this technique, the plants have acquired most of their mutations through mutagenesis treatment, and as a result, there are no obvious differences to be seen. Line segregation and homozygosity of the plant for the demonstration of mutant phenotypic variations in the following generation (M2) produce the F1 population by crossing the wild-type variety with the mutant phenotype and further F1 populations are crossed to produce the next generation (F2). Next-generation sequencing (NGS) is then used to analyze the sequencing data from individuals with a wild-type and mutant phenotype to identify mutations that are linked to the mutant phenotype. It characterizes the mutation by evaluating SNP frequency in the wild-type DNA and mutant offspring of the M3 generation acquired just after self-pollination of the M2 heterozygous lines, as shown in Figure 1. The technique was initially applied on rice by [67]. When coupled with MutMap, a similar approach called MutMap-GAP facilitates the recognition of the specific gene from gaps within the specific reference genome. MutMap has been used to determine regions that may have the desired mutation. The de novo assembly is carried out after

determining the targeted region that is preceded by alignment and detecting the mutation from different regions [68]. Not only did the mutation and mutagenesis methods allow us to discover genetic regions associated with the desired characteristics, but they have also contributed to the expansion of diversity and the development of mutant rice varieties with improved agronomic quality.

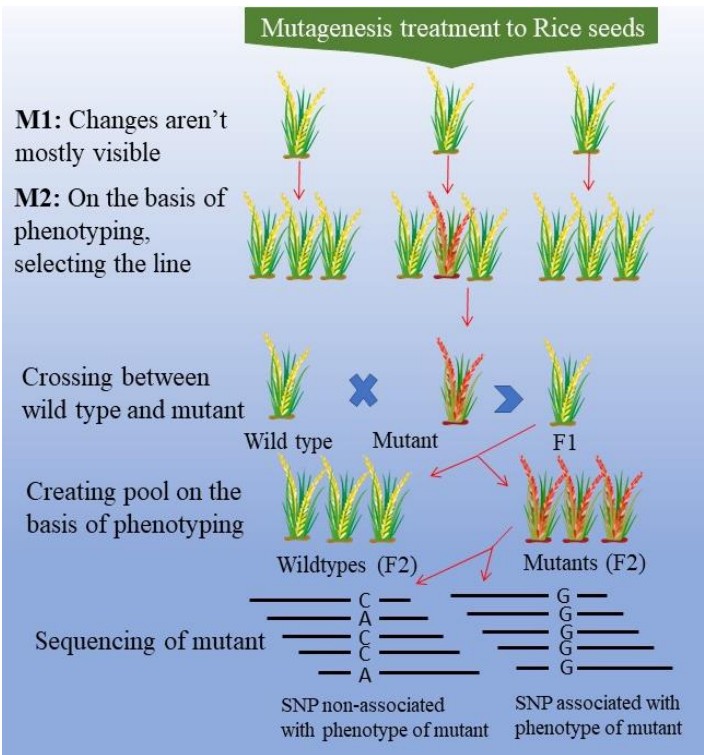

**Figure 1.** The process of MutMap approach.

## 6. Integrative Omics Technologies for Enhancement of Rice Nutritional Quality Traits

Extensive understanding of agronomically relevant characteristics is facilitated by integrating effective studies that incorporate relevant information from proteomics, metabolomics, genomes, nutrigenomics, and transcriptomics. Meanwhile, many advancements in advanced technology are made in sequencing methods and data analysis, as well as the availability of the entire rice genome sequences, have accelerated attempts to enhance significant characteristics of crops nutritionally and agronomically [12]. Along with identifying genes involved in agronomic characteristics, an integrated omics-based research method attempts to unravel biochemical mechanisms and connections between biomolecular regulation and activities. Although the whole rice genome was sequenced a long time ago, several specific proteome, transcriptome, ionome, and metabolome studies have been conducted to improve the nutrient content of grains by using genomic data. However, still, there are large embedded omics initiatives that have not been used to their full capacity [69].

Effective gene information exploration requires a thorough knowledge of genetic control, molecular mechanisms, complicated gene environments and gene-with-gene interactions. Omics technologies enable the collection of the comprehensive data necessary for product development [70]. Numerous genes of rice influencing nutrient quality-related characteristics are being actively researched in new ongoing studies. The knowledge on these genes is growing due to various omics tools and being effectively investigated using molecular breeding, transgenic approaches, and even comparatively new technologies such as genome editing [71].

Among the several combined omics studies providing a large amount of data includes research in which researchers found 3000 metabolites by using metabolomics from ten cooked-rice varieties. Functional genomics are utilized to explore the genetic differences

that result in metabolite diversity. It also allowed researchers to investigate the gene variety of phenolic chemicals and identify SNPs in their UTRs, which control gene expression [12]. An investigation was performed to examine the genetic and metabolic differences between traditional and enhanced waxy rice cultivars. This study determined nutritional and yield variations between three varieties of rice [72]. There is a scarcity of such integrated omics initiatives aimed at deducing connections between varieties and extensively studying rice crops' nutritionally and agronomically significant characteristics. Due to the significance of CREs for gene expression in plants, further study is required to investigate CRE regulation mechanisms and metabolic links. Thus, it is crucial to analyse the CREs associated with crop quality and their transcriptional and translational regulatory changes. It is believed that the integration of CREs and genome editing approaches would make it possible to manipulate numerous features in rice simultaneously (Figure 2).

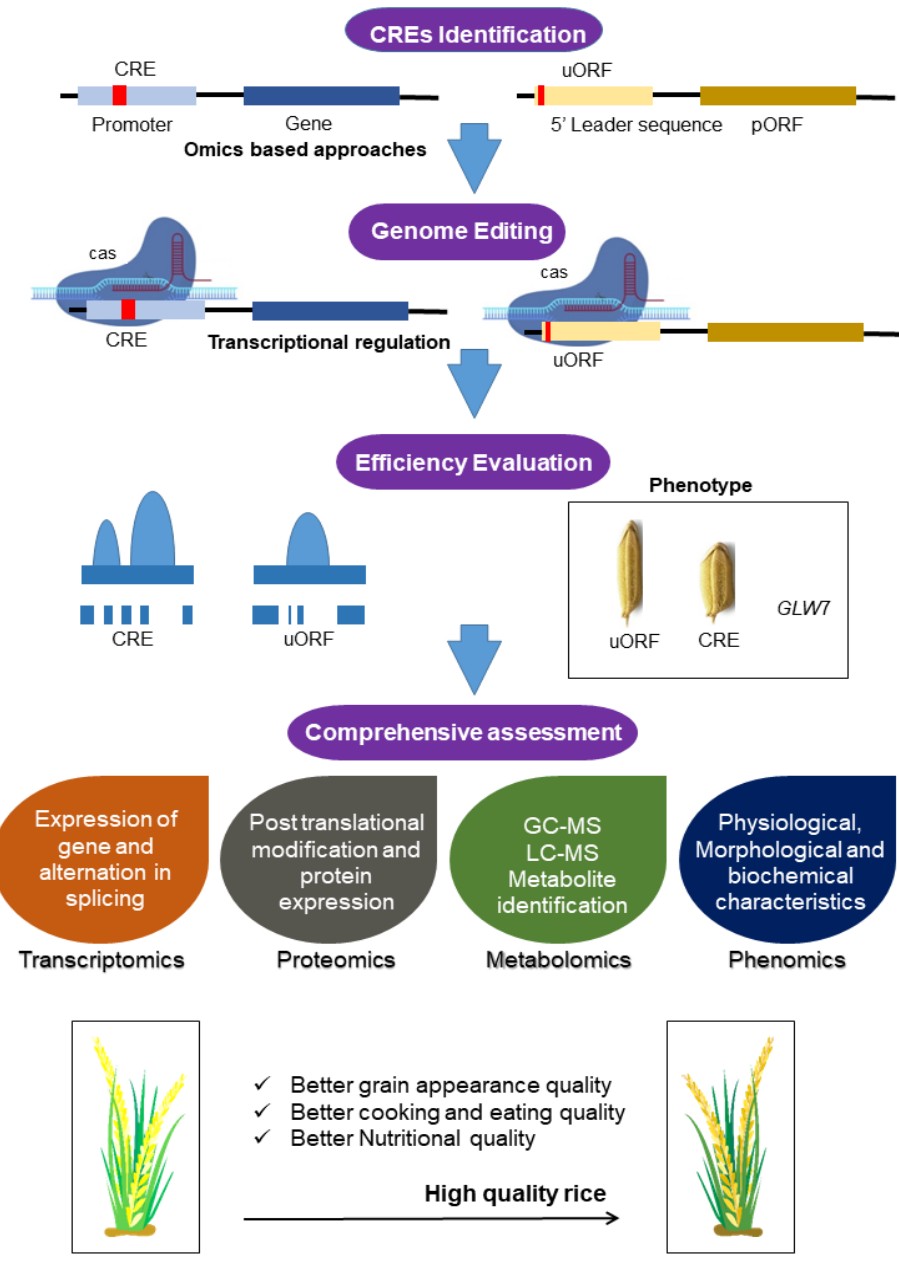

**Figure 2.** Flowchart demonstrating the use of genome editing techniques to produce high-quality rice grains.

### 6.1. Genomics and Pan-Genomics Analysis

Several rice genomes have been constructed to diverse levels of quality during the last two decades, ranging from draft genome to relatively close high-reference sequencing, using a variety of sequence and construction methodologies [67]. Using single molecule real-time sequencing (PacBio, Menlo Park, CA, USA) and high-throughput short-read sequencing (Illumina, San Diego, CA, USA), several additional rice genome sequences have been created since the availability of the nip reference sequence from the International Rice Genome Sequencing Project in 2005 [73] based on bacterial artificial chromosome Sanger sequencing. BioNano genome mapping (BioNano Genomics, San Diego, CA, USA), Single-molecule sequencing, and high-throughput chromosomal conformation capture (Hi-C) sequencing are some of the latest genome assembling methods that make use of the enhanced read length and/or base precision of single-molecule sequencing. Combining these techniques, as well as the development of novel assembly tools, has resulted in a significant improvement in assembly quality, allowing for the generation of highly consistent chromosome-level assemblages at a low cost. PacBio's introduction of HiFi sequencing, in 2019, constitutes a technological milestone [74]. Since the introduction of R498, several rising rice genomes have been constructed, such as 12 genomes with an average gap count of 18 per genome, an average completeness of up to 98.75 percent [75], and 31 genomes with an average contig N50 of 12.89 Mb and a gap count of 63 [76]. Through precise gene mapping and genome-wide association studies (GWAS) [77], these genomes have aided in the analysis of population diversification and gene characterization.

The BB genome of *Oryza punctata* [78], the KKLL genome of *Oryza coarctata* [79], and six AA genomes have been constructed in the past decade using different methodologies. All wild rice genomes are able to be arranged with the conjunction of SMRT, PacBio sequence analysis, BioNano genome mapping, and Hi-C sequence analysis, as demonstrated by the last several arrangements of a rising chromosome-scale AA genome (contig N50, 13.2 Mb) from an extremely heterozygous *Oryza rufipogon* accession [80] and of the heterozygous allotetraploid *Oryza alta* CCDD genome (contig N50, 18.2 Mb) with comprehensiveness similar to that of the cultivated rice genome.

Numerous sequences and genes that are not included in the nip reference genome have been found via pan-genomic investigations, as well as genes that are not present across all genomes. It was found that 1300 novel genes (missing from nip) and 3144 dispensable genes, including several genes involved with disease resistance, were found in the draught genomes of Xian IR64 and aus DJ123 when compared to the reference genome. It was found that 268 mb of novel sequences, 12,465 complete new genes, and 19,721 nonessential genes were discovered in a study of the 3010 rice genomes [81]. Researchers discovered 10,872 novel genes that were at least largely absent in nip and 16,208 expendable genes using de novo assembly drafted genomes of 66 typical cultivated and wild A-genome rice accessions [82]. For agronomic variables such as grain length, grain breadth, and bacterial blight- resistant rice, pangenomic studies have helped improve gene mapping by GWAS and offered fresh information on rice's evolution and domestication [81,82]. Technology advancements in genomics and synthetic biology have the potential to speed up agricultural improvement [83]. It is possible to create rice cultivars with high yield, excellent quality, and resistance to stressors by precise breeding of numerous favorable alleles [84]. Many genes in Asian rice have been altered to increase quality, resilience, or yield, in contrast to the de novo adoption of *Oryza alta* [85]. Using these technologies on a wide scale for rice development demands an in-depth awareness of the agronomic characteristics' complicated genetic architecture. As it is, the bulk of genetic variation in rice is not covered by existing rice pan-genomes, which include just AA genomes from a few Oryza species. For this reason, it is imperative to build Oryza pangenomes to incorporate different cultivated and wild rice genomes, so that the entire Oryza genus may be studied.

*6.2. Transcriptomics: Rice Nutritional Quality Enhancement through RNA Sequences*

Transcriptomics is an analysis of RNA expression patterns that considers both coding and noncoding sequences of RNA present in cells at any particular point. Different methods have been established to characterize the pattern of gene expression of rice plants, including the sequences of RNA and microarrays. Many RNA sequence samples are uploaded in the NCBI SRA database, and this database increases day by day with RNA sequences [86]. Likewise, using the RNA sequence data, the gene expression omnibus (GEO), has been openly available for the usage of collected data of transcriptome profiling by using microarrays (GEO) (https://www.ncbi.nlm.nih.gov/geo/ (accessed on 10 March 2022)). The first attempt was made in the 1990s to identify the whole transcriptome. The transcription patterns of the whole genomes of rice species (*O. sativa* indica and japonica) were acquired by RNA sequencing [87]. The researchers found that the analysis of RNA-seq on a large scale improved the coding of the rice genome by identifying 101 novel loci and 1584 unique peptides that are matched with new peptides. Additionally, different splicing has been examined concerning the regulation of mineral nutrient equilibrium in rice [88]. Transcriptomic and microarray investigations have been undertaken to better understand the antagonistic relationships between micronutrients [86]. [89] used microarray data to determine the antagonistic relationship between P and Fe in rice plants. Microarray analysis of rice roots was used to characterize relationships based on molecular genetics for adjustment of macronutrient (nitrogen, phosphorus, potassium) deficiencies.

Transcriptomics permits the investigation of changes in gene expression, the explanation of previously unknown genes, and the control of genes. Large-scale genome-wide association studies (GWAS) and transcriptome studies have assisted in the prediction of genes that affect the glycemic index (GI) of rice and less glycemic index in nutritional quantity is critical for Type II diabetes patients and some dietary diseases including hypertension and diabetes [90]. To generate a cell-type transcriptome database, laser microdissection was employed as well as microarray profiling [91]. The majority of rice transcriptome profiling research has focused on stress tolerance with comparatively little effort required for nutritional quality traits.

*6.3. Proteomics: Rice Nutritional Quality Enhancement through Protein*

Proteomics is the study of a large number of proteins in an organism, their location, quantity, and posttranslational alterations. Proteomics supports genomics and transcriptomics to enhance our understanding of molecular structure and function. Numerous advanced techniques, including gel-free techniques such as MALDI-TOF, tandem mass spectrometry (MS), liquid chromatography mass spectrometry (LCMS-MS), and gel-assisted techniques. Researchers have used combinations of these technologies to develop large amounts of proteomics information [92]. Proteomics is concerned with the pattern of translation of biochemical and physiological activities in rice plants. Several studies demonstrate the understanding of different levels of expression of bioactive chemical compounds providing a more in-depth examination of rice's nutritional quality under a variety of circumstances. The comparison of KDML105 and Mali Daeng (MD) rice revealed variations in the expression trends of antioxidant activity, anthocyanins, phenolic compounds, and during germination. The research reveals that red rice MD had more antioxidant activity, anthocyanin, and phenolic compounds when compared to KDML105 white rice [93]. Proteomics techniques appear to be promising for assessing the potential influence of transgenic on the nutritional content of food or any expression of genes followed by translation functions [92].

Many investigations focusing on rice seed storage protein expression levels and their relationship with nutritional quality have been conducted to better understand protein regulation. Proteomic techniques were utilized to characterize 302 candidate proteins for their biochemical functions such as catalytic and hydrophilic activities, as well as binding affinity in metabolic pathways [94]. Proteome and glycomic analysis of chalky rice grains exposed to high environmental stresses showed the breakdown of starch rather than the

synthesis of starch that is responsible for rice chalkiness. Proteome analysis was performed on 25 genes that are related to metals in rice and includes zinc and iron concentration in seed which are beneficial for biofortification reasons. The finding focuses on the chemical process through which metal is transported from flag leaves to seeds [95]. The two cultivar-specific high-yielding rice cultivars were analyzed to determine the phenolic content and antioxidants when exposed to varied ozone concentrations. The study observed alterations in the antioxidant defense pathways and proteome of the leaf, as well as a decline in grain quality and production [96]. Thus, the proteome study is integrated with genetics to aid in understanding the changing protein content in plants and their genes involved in the efficient protein concentration in grains.

*6.4. Metabolomics: Rice Nutritional Quality Improvement through Metabolic Regulation*

Metabolomics is the qualitative and quantitative analysis, systematic identification, and quantification of small molecules in biological organisms. Many such analytical methods for the analysis of plant metabolomes include mass spectrometry (MS) techniques such as liquid chromatography (LC-MS), gas chromatography (GC-MS), capillary electrophoresis (CE), nuclear magnetic resonance (NMR), metabolic fingerprinting using ion cyclotron spatial mass spectrometry (FT-MS) and Raman spectroscopy (microscopy) [97]. Metabolomic studies found variations in bioactive chemicals between uncooked and cooked rice varieties. The research identified thousands of chemical compounds and gene SNPs controlling nutritionally significant metabolism. The variability in the metabolome of cooked rice was investigated, as well as the influence of SNPs on several cultivars of rice that have nutrient content such as vitamin E and phenolics concentration [98]. Metabolomics is a term that refers to the molecular phenotyping of biological activities and metabolic processes that occur inside it. The technique was applied to 68 rice accessions for metabolic phenotyping and identified 10 typical metabolites. In this research, metabolite profiling of rice is introduced and utilized to determine the genes, QTL, and modifications that are the main reason for nutrient quality in rice grains [99]. The research examined the rice metabolome genetically and identified 2800 QTLs for 900 metabolites. This research illustrates the twenty-four candidate genes that are primarily responsible for the amount of rice phenolic chemicals [100]. Extensive studies are establishing the usefulness of metabolomics in elucidating the biomolecular mechanisms behind various quality-related characteristics. Additionally, researchers have developed a multiplatform metabolomics technique to analyze various metabolomics data sets to identify discriminating chemical compounds important to the nutritional quality in rice. Metabolomics enables the rapid evaluation of a large number of metabolites and identifies several genetic architectures that are responsible for the regulation of bioactive chemical compounds on the nutritional content of rice. Metabolomics data can assist in the identification of breeding material for superior rice variety development [98].

Many studies have revealed that metabolomics-specified breeding is an important tool for improving the genetics of rice crops. The process of metabolite profiling is used to determine the vitamins, secondary metabolites, amino acids, and cofactors to help in enhancing the existing information that is supplied by dietary supplementation. Research has revealed that metabolic phenotype is correlated with the geographical origins of Japonica and Indica rice varieties [101]. Along with the metabolomics studies of commercial rice, comparable research has been conducted on wild rice varieties to identify and develop useful food. For example, the North American (*Zizania palustris*) and Chinese (*Zizania latifolia*) species vary in 357 secondary metabolites contents, mostly in catechins and anthocyanins [102]. Likewise, metabolomics research and analysis to determine the beneficial chemicals present in differently shaped embryos (large and regular size) of rice grains has revealed that the large vacuoles may have a maximum level of accumulating beneficial chemicals, indicating the maximum nutritional grain quality of rice [103]. Metabolomics has permitted a thorough examination of the micro-metabolites of rice plants that are intimately associated

with phenotypic characteristics. The primary difficulty in metabolomics technology is the interpretation and extraction of massive amounts of data from biological systems.

*6.5. Nutrigenomics Approach in Rice*

Nutrigenomics emphasizes the connection between gene expression and food constituent. Its objective is to obtain a molecular knowledge of how dietary regimens and nutrients affect gene expression. Malnutrition is mostly caused by an acute deficiency in vitamins and minerals. Nutrigenomics seek to increase nutritional food quality by increasing the bioavailability of macronutrients and micronutrients in cereals and vegetables or by adding bioactive chemicals into agricultural crops [104]. Additionally, the functional characteristics and nutrigenomic impacts of germinated brown rice rich in bioactive compounds have been investigated to obtain a better understanding of the role of grain in balanced diets. Applying gene-based markers and modern technology is helpful for breeders to accumulate alleles of genes known to influence nutritional grain quality characteristics in rice [105]. Recently, great success has been achieved in amino acid and grain protein content, glycemic index, vitamins, phytic acid, flavonoid and phenolic compounds, iron and zinc and iron content as well as linkage with QTLs, but more studies and efforts are needed to enhance the nutritional quality of rice and its curative properties [5]. Rice improvement has been shown in recent studies with the introduction of high-protein and zinc-rich rice cultivars, which ultimately enhances the nutritional value [95].

## 7. Conclusions

Rice innovation has been largely concentrated on yield-related attributes, with less concentration on nutritional quality enhancement. Due to the rising population and limited resources, the use of advanced technologies and protocols will be needed to increase crop quality. In comparison to other omics branches, transcriptomics and genomics have made substantial progress, and the ensuing combination of genomics and transcriptomics has grown more common. Numerous GWAS and QTL mappings have been conducted in recent times for commercial quality-related characteristics such as elongation ratio, grain size and fragrance yield, but these efforts rarely achieve the best nutritional quality of rice. The resources created for GWAS and QTL analysis may be effectively used to construct genetic selection and prediction models. The sequential use of many techniques that utilize comparable resources will be a successful approach for crop enhancement initiatives. The combination of high-genotyping methods with genome selection (GS), QTL mapping, and GWAS appears to be viable and cost effective. Therefore, the publicly available omics resources for rice must be efficiently examined. Mutagenesis is one such field that has benefited from the advancement of NGS technology. The new mutation mapping techniques are more accurate, cost effective and rapid. Similar techniques investigating the effectiveness of diverse tools and methodologies are anticipated. Apart from making significant achievements in other omics fields, interdisciplinary research and integrated techniques have not been fully used to achieve targeted rice grain quality. In addition to conventional breeding, the omics strategy is shown to be more effective in improving traits. Integrating omics techniques such as proteomics, genome, ionomics, transcriptomics, and metabolomics is critical for obtaining a full picture of rice's nutritional quality-related characteristics. The critical evaluation of rice improvement advances has revealed a shortage of cheap and practical elevated phenotyping platforms capable of integrating with many other branches of omics for effective research. The implementation of omics is improving the nutritional content of rice, and it can aid in the eradication of hidden hunger and help to achieve the sustainable development goals of the United Nations organization for the upcoming World.

**Author Contributions:** M.J.Z. wrote the manuscript. S.T. designed and guided with an innovative ideas. S.T. and K.A. revised and helped in preparing the manuscript. All authors have read and agreed to the published version of the manuscript.

**Funding:** This work was supported by the National Natural Science Foundation (U19A2025 and 31870229), the Strategic Priority Research Program of the Chinese Academy of Sciences (Grant No. XDA24010404).

**Institutional Review Board Statement:** Not Applicable.

**Informed Consent Statement:** Not Applicable.

**Conflicts of Interest:** The authors declare no conflict of interest.

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
