# Peer review of "Integrated Genetic and Omics Approaches for the Regulation of Nutritional Activities in Rice (Oryza sativa L.)"

_agriculture, doi:10.3390/agriculture12111757_

Round 1

Reviewer 1 Report

Comments are provide in the attached manuscript.

Major Comments:

1. List of nutrient-enriched rice varieties developed and released may provided in the tabular form.

2. Section 3 & 4 dealt with same topic & hence merge them under single heading. 

Reviewer 2 Report

Authors of this manuscript dealt with presentation of a review on the genetic and omics approaches for the regulation of nutritional activities in rice. The topic is of interest, but the synthesis of the review, treatment of various aspects and discussion are not very well presented. Author have to re-work on their manuscript and improve the overall manuscript for good readability (especially the language) with updated information. In many places in the text, take-home-message is missing and only limitation is projected. Title needs a change since nutritional activities' doesnot convey the total quality attributes.

My comments are:

1. Lack of  connectivity among the presented information. 

For ex. Line 83 authors mentioned 'Golden rice has encountered difficulties associated with the correct management of Genetically Modified Organisms (GMOs) and opposition to GM technology [10].'  Here, there is a bit of contradiction (encountered difficulties with correct management). The cited reference is of 2005, and authors should be aware that much has changed in the past 15-17 years. 

Line 86: ‘and ultimately cause the researchers to abandon transgenic technologies. Several of these technologies, collectively referred to as genome  editing technologies, have demonstrated enormous value’. Here again, there is a mis-information from authors that ‘researchers to abandon transgenic technologies’; in reality, this is not true and many countries have trying to adopt GM crops. The contradiction can be seen again in the next sentences.  Referencing has to be updated since citing 2013 reference for genome editing has no meaning.

2.Line 101: very strange sentence ‘We have discussed the efforts taken in this article to increase the nutritional content of rice.

3. QTLs for grain Fe and Zn in rice has been well studied and there are several report in the past ten years.  I think authors should discuss more on this with relevant studies and not only meta-analysis studies.

4.Line 149-151: Its not clear what authors mean by ‘combined or antagonistic’ in this sentence.  ‘Although research on the combined or antagonistic impact of multiple QTLs for enhanced nutritional characteristics has been done, further studies are necessary to explore the combined or antagonistic impacts of multiple QTLs for enhancing nutritional contents

5.References in Table 1 are old, should be updated with recent literature.

6.Section on ‘Efficacy and capability of nutrient-rich rice breeding through genome selection (GS) should be centered around nutritional quality.

7.Section on ‘Mutation mapping and mutagenesis techniques: Triggers the nutritional quality of rice-its not clear what triggers nutritional quality, by the title it looks the techniques trigger the quality. Strange title. Please change. The writing in this section should be centered around nutritional quality. Authors can also refer to Mut-map+ which doesnot require crossing of wild type variety with the mutant.

line 240-263 should have more information on the results obtained in terms of data and interpretation rather than describing the techniques.

8. Figure 1 is very simple in drawing. I feel more inputs into how the science behind the techniques will play role in nutritional quality should be projected.

9. Line 456- authors used the term  Nutrigenomics .  The definition of Nutrigenomics  is the study of the effects of food and food constituents on gene expression, and how genetic variations affect the nutritional environment”.  This section lacks good discussion and authors only mentioned what has been presented already in other sections. Rewriting is required to convey as per the title.

Round 2

Reviewer 2 Report

Authors have done good (to some extent) revision. However there are still some concerns.

1. I feel language still needs to be checked. for ex. line 665- sentences like The transgenic techniques are allowed to improve grain nutrition at a quicker rate. Genomic technologies have the potential to improve rice nutritional quality by working hand in hand with genetic improvement.

Line 641-large embryos may have maximum accumulating beneficial chemicals

2. I suggest authors to write more technical details in to Table 1. Some Rice genes that control regulation of nutritional quality traits. Presently it looks very general.

3.   I suggest authors to re-write section, 5. Mutation mapping and mutagenesis techniques. Its not clear what they want to convey. There is seems to be little relevance of the Mut-Map to nutritional quality. This needs to be checked and modified.

4. Thorough checking of the entire manuscript is required.

5. Table 1 may be expanded to include other QTLs for nutrient enhancement in rice for ex. Amino acid content, lysine, P, K, Mg, Boron, Zn Fe.

6. Section 4 needs to have more discussion on   related to nutritional quality.  Few references can be useful (Descalsota-Empleo GI, Noraziyah AAS, Navea IP, Chung C, Dwiyanti MS, Labios RJD, Ikmal AM, Juanillas VM, Inabangan-Asilo MA, Amparado A, Reinke R, Cruz CMV, Chin JH, Swamy BPM. Genetic Dissection of Grain Nutritional Traits and Leaf Blight Resistance in Rice. Genes (Basel). 2019 Jan 8;10(1):30. doi: 10.3390/genes10010030); 

Bollinedi H, Yadav AK, Vinod KK, Gopala Krishnan S, Bhowmick PK, Nagarajan M, Neeraja CN, Ellur RK, Singh AK. Genome-Wide Association Study Reveals Novel Marker-Trait Associations (MTAs) Governing the Localization of Fe and Zn in the Rice Grain. Front Genet. 2020 Apr 22;11:213. doi: 10.3389/fgene.2020.00213.). 

Last line in this section needs to have more updated citation (Patra, B., Majhi, P. K., Tripathy, S. K., Tripathy, S. P., Khan, A., Behera, P. P., Das, S., & Ahamad, A. (2022). Genomic-Assisted Breeding Tools for Grain and Nutritional Quality Improvement in Rice. International Journal of Environment and Climate Change12(1), 10-24.; Bartholomé, J., Prakash, P.T., Cobb, J.N. (2022). Genomic Prediction: Progress and Perspectives for Rice Improvement. In: Ahmadi, N., Bartholomé, J. (eds) Genomic Prediction of Complex Traits. Methods in Molecular Biology, vol 2467. Humana, New York, NY. https://doi.org/10.1007/978-1-0716-2205-6_21). 

7. There is also a need for discussing in depth on Mutation mapping and mutagenesis techniques: Triggers the Impact on nutritional quality of rice
